# Targeting of the Mitochondrial TET1 Protein by Pyrrolo[3,2-*b*]pyrrole Chelators

**DOI:** 10.3390/ijms231810850

**Published:** 2022-09-16

**Authors:** Veronika Antonyová, Ameneh Tatar, Tereza Brogyányi, Zdeněk Kejík, Robert Kaplánek, Fréderic Vellieux, Nikita Abramenko, Alla Sinica, Jan Hajduch, Petr Novotný, Bettie Sue Masters, Pavel Martásek, Milan Jakubek

**Affiliations:** 1Department of Paediatrics and Inherited Metabolic Disorders, First Faculty of Medicine, Charles University and General University Hospital in Prague, Ke Karlovu 455/2, 128 08 Prague, Czech Republic; 2BIOCEV, First Faculty of Medicine, Charles University, 252 20 Vestec, Czech Republic; 3Department of Analytical Chemistry, Faculty of Chemical Engineering, University of Chemistry and Technology, 166 28 Prague, Czech Republic; 4Institute of Pathological Physiology, First Faculty of Medicine, Charles University, 128 53 Prague, Czech Republic; 5Duke University Medical Center, Department of Biochemistry, Durham, NC 27707, USA

**Keywords:** TET1 protein inhibitor, pyrrolo[3,2-*b*]pyrrole, hydrazone, mitochondria

## Abstract

Targeting of epigenetic mechanisms, such as the hydroxymethylation of DNA, has been intensively studied, with respect to the treatment of many serious pathologies, including oncological disorders. Recent studies demonstrated that promising therapeutic strategies could potentially be based on the inhibition of the TET1 protein (ten-eleven translocation methylcytosine dioxygenase 1) by specific iron chelators. Therefore, in the present work, we prepared a series of pyrrolopyrrole derivatives with hydrazide (**1**) or hydrazone (**2**–**6**) iron-binding groups. As a result, we determined that the basic pyrrolo[3,2-*b*]pyrrole derivative **1** was a strong inhibitor of the TET1 protein (IC_50_ = 1.33 μM), supported by microscale thermophoresis and molecular docking. Pyrrolo[3,2-*b*]pyrroles **2**–**6**, bearing substituted 2-hydroxybenzylidene moieties, displayed no significant inhibitory activity. In addition, in vitro studies demonstrated that derivative **1** exhibits potent anticancer activity and an exclusive mitochondrial localization, confirmed by Pearson’s correlation coefficient of 0.92.

## 1. Introduction

Epigenetics involves any process modifying the activity of genes without permanently changing the DNA sequence; these changes are further inherited to daughter cells [1]. DNA and histone modifications, nucleosome repositioning, higher-order chromatin remodeling, noncoding RNAs, and RNA or DNA editing are epigenetic mechanisms responsible for the development of epigenetic alterations [2]. Abnormal expression patterns and genomic changes in chromatin regulators can have serious consequences and may lead to the development and poor prognosis of various cancers [3].

The first and best studied covalent DNA modification that has been described is the methylation at the fifth position of cytosine [4]. CpG island methylation is an important epigenetic mechanism regulating gene expression, genome stability, and genomic imprinting [5,6].

In 2009, 5-hydroxymethylcytosine (5hmC) was first identified as a new cytosine modification. It was shown that 5hmC is generated by 5-methylcytosine (5mC) oxidation through the ten-eleven translocation (TET1/2/3) family of enzymes, which further oxidize 5hmC to 5-formylcytosine (5fC) and 5-carboxylcytosine (5caC) [7,8]. 5fC and 5caC are then converted back to unmodified cytosine by thymine-DNA glycosylase and by the base excision repair mechanism [9,10]. In addition to the activity of TET enzymes in active demethylation, TET1 acts both as a transcriptional coactivator and corepressor, in which its transcriptional activity is either dependent upon or independent of its demethylase activity [11,12,13]. All three TET1 enzymes are Fe(II) and 2-oxoglutarate (2OG)-dependent dioxygenases, each of which contains a highly conserved carboxy-terminal catalytic region. This region is composed of a cysteine-rich (Cys-rich) region and a double-stranded β-helix (DSBH) domain [7,14]. Key catalytic residues that bind Fe(II) and 2OG are present within the DSBH domain [15].

Despite studies designating TET1 as a tumor suppressor, there is evidence that links global hypomethylation and levels of 5hmC to a variety of neurodegenerative and oncological disorders [16,17]. Huang et al. showed that TET1 is a direct target of mixed-lineage leukemia (MLL)-fusion proteins and is upregulated in MLL-rearranged leukemia, which leads to a global increase in 5-hydroxymethylcytosine levels [18]. The oncogenic role of TET1 lies in coordination with MLL-fusion proteins in regulating their critical cotargets. In studies involving the level of methylation in patients with triple-negative breast cancer (TNBC), significant overexpression of the TET1 protein [19] was demonstrated. Increased levels of TET1 were associated with TNBC-specific hypomethylation of up to 10% of queried CpG sites and a lessened overall survival of patients [19]. Similarly, Zhang et al. reported that promoter hypomethylation activated an expression of oncogenes in patients with ovarian cancer [20]. Moreover, in a study by Chen et al., high levels of TET1 were observed and contributed to the hydroxymethyl-dependent activation of oncogenic pathways, including the regulation of CK2α [21]. Takai et al. found high levels of 5hmC and the TET1 protein in samples of proneural glioblastomas. In this work, the authors proved that the production of 5hmC is required for the tumorigenicity of glioblastoma cells [22]. In another study, the authors found that TET1 gene expression is elevated in non-small-cell lung carcinoma (NSCLC) cell lines and contributes to cell growth. TET1 was identified as an oncogene in NSCLC, whose gain of function following the loss of p53 may be exploited by targeted-therapy-induced senescence [23]. Authors of this cited study suggested that the development of small-molecule inhibitors of TET1 in concomitant use with genotoxic drugs may contribute to a more effective therapeutic strategy for patients with lung cancer carrying the p53 mutation. Similarly, it was found that circular RNA circ_0007919 inhibits CRC cell growth and migration via regulating the miR-942-5p/TET1 axis [24].

Nevertheless, the above discusses only the role of 5hmC in the context of the nuclear DNA and/or TET1 protein levels/activity in oncogenesis, but mitochondrial epigenetic modifications are neglected. Although mitochondrial epigenetics has been intensively studied, especially in recent years, our knowledge is limited [25]. High similarity between nuclear and mitochondrial epigenetic mechanisms was assumed, except for histone modifications (mitochondria did not contain histones). However, the presence of modified bases, such as 5mC and 5hmC in the mitochondrial DNA, was confirmed [26]. In the case of the TET1 protein itself, the available information is unfortunately much more limited. On the other hand, its mitochondrial presence was reported in some high-impact studies [27,28,29]. Dzitoyeva et al. reported that valproic acid (an intensively studied anticancer agent) [30] decreases the 5hmC/5mC ratio in both nuclear and mitochondrial DNA and TET1 protein levels in mouse 3T3-L1 cells [27].

The above conclusions imply that targeting both the nuclear and mitochondrial TET1 protein could be promising therapeutic modalities in cancer treatment. An interesting clue that epigenetic agents are well suited for a combination with classically used therapeutic methods [31] derives from the fact that the oxygenase activity of TET enzymes is highly dependent on the presence of Fe(II) and 2-OG and that key catalytic residues that bind Fe(II) and 2-OG are present within the DSBH domain [15]. This indicates an important role for Fe(II) chelators in targeting TET1 and thereby its inhibition. In the our last works, we observed that Fe(II) chelators can display potent inhibition activity against the TET1 protein [32,33].

The chelator core of the ligands was prepared by the procedure of Gryco et al. in a single-step reaction [34]. Hydrazones are effective metal-chelating groups. They are also important pharmacophores [35,36,37,38,39,40]. Thus, the diester of tetraaryl pyrrolo[3,2-*b*]pyrrole was converted to a corresponding hydrazide, which was then treated with a series of 2-hydroxybenzaldehydes via Schiff base condensation to obtain the hydrazone derivatives, which could then serve as chelating units. Recently, we described 2-hydroxybenzylidene and 2-N-heteroarylidene hydrazone derivatives with benzoisothiazole-1,1-dioxide and cholic acid moieties that display a facile interaction with iron metals, in concert with an inhibitory effect towards TET1 [33]. A similar approach for the synthesis of Fe(II) chelators was also described by Zhu et al. [41]. A promising structural motif for the design of Fe(II) chelators with a possible inhibitory activity on TET1 could be based on fluorescent pyrrolo[3,2-*b*]pyrrole as a chelator core. Its derivatives show affinity for the transition metal ions [42] and enable an effective substitution by binding groups for the Fe(II) ion. In the present study, we report new hydrazide and hydrazone chelators, which show instantaneous responsiveness towards iron cations with facile operation and high sensitivity as a novel class of iron chelators and potential TET1 protein inhibitors.

## 2. Results and Discussion

### 2.1. Synthesis of Compounds ***1*****–*****7***

Recent studies strongly imply that iron chelators represent promising structural motifs for targeting the TET1 protein [32,33,43]. Gangadharam et al. reported that hydrazide derivatives are potent inhibitors of iron oxidoreductases [44]. Our latest work shows that aromatic hydrazones display promising inhibition activity against the TET1 protein. Molecular docking studies indicate that aromatic interactions could play a significant part in the inhibitor binding to the enzyme active site [32,45]. Pyrrolo-pyrroles, such as pyrrolo[3,2-*b*]pyrrole, are infrequently used scaffolds in the synthesis of enzyme inhibitors; nevertheless, as chelating agents, their application could offer some promising benefits. They are electron rich and thereby could support the chelation of iron [46]. They can effectively interact via π−π and CN···H interactions [47]. They are strong fluorophores and enable observed inhibitor distribution in cells [48]. TET1 proteins are localized in the nucleus and mitochondria [27,49], and the intracellular distribution of inhibitors can strongly influence their utility. Thus, we decided to combine a strongly fluorescent pyrrolo[3,2-*b*]pyrrole core (fluorophore for intracellular visualization) with a suitable metal-chelating unit (a bioactive part responsible for the inhibition of the TET1 enzyme) (Figure 1). We suggest that preparations of tetraaryl pyrrolo[3,2-*b*]pyrrole dihydrazide and its subsequently substituted products could be promising tools for the construction of a new type of TET1 protein inhibitors.

The fluorescent core of the potential iron chelators was prepared in moderate yields in a single-step reaction by condensation of methyl 4-formylbenzoate, 4-methylaniline, and 2,3-butanedione in the presence of *p*-toluenesulfonic acid and acetic acid at 90 °C for 3 h, following the procedure outlined by the research group of Gryco [34]. The yellowish tetraaryl pyrrolo[3,2-*b*]pyrrole compound **7** was converted to hydrazide by treatment of ester groups with hydrazine hydrate in the presence of 10 mol% 4-dimethylaminopyridine at 90 °C for 3 days to obtain highly fluorescent tetraaryl pyrrolo[3,2-*b*]pyrrole dihydrazide **1**. Compounds **2**–**6** were then prepared by reacting hydrazide **1** with substituted 2-hydroxybenzaldehydes in ethyl alcohol in the presence of a catalytic amount of acetic acid overnight at room temperature (Figure 1). This resulted in yields of 50–70% (Figure 1). ^1^H and ^13^CNMR (DMSO-d6) and HRMS spectra of compounds **1**–**7** are shown in Appendix A Appendix A, respectively. Compound **1** represents hydrazide, which possesses an O–N bidentate chelating unit, while ligands **2**–**6** represent aroylhydrazones, which possess tridentate O–N–O chelating units. The effects of synthesized hydrazides and hydrazones on the inhibition of the TET1 protein were subsequently investigated.

### 2.2. FTIR and Raman Analysis of Compounds ***1***–***7***

The core of pyrrolo[3,2-*b*]pyrrole, surrounded by four *para*-disubstituted benzene moieties, is the central part of the molecular structures of compounds **1**–**7**. Thus, all these compounds contain similar structural elements, so it is expected that the vibrational spectra will have some similar spectral regions and bands. Indeed, the obtained Raman spectra (Figure 2a) in all cases consisted of a number of distinct bands with very similar positions. The bands at 1605–1609, 1450–1446, and 1551–1556 cm^−1^ were assigned to C=C symmetric stretching of the pyrrolo[3,2-*b*]pyrrole core and benzene ring, whereas the in-plane bending vibrations of the pyrrole ring were located at 1490–1497 and 1403–1405 cm^−1^ [50,51]. The regions at 1285–1275, 1190–1186, and 1014–1019 cm^−1^ were associated with C–N stretching and CH and NH bending vibrations, respectively [51]. Similar vibrational bands appeared in the same regions of the FTIR spectra (Figure 2b). In addition, characteristic IR bands of hydrazide vibrations (C=O stretching and NH bending) were found at 1655–1672 cm^−1^. In compound **1**, two oppositely located *para*-disubstituted benzene rings carry peripheral hydrazide groups, which in compounds **2**–**6** serve to link different substituents. The precise position of vibrational bands depended on the nature of these substituents, the hydrogen bonding between the CO and NH groups in hydrazides, and the conformation of these moieties, specifically to the torsional angles around C–N and C–C bonds. Free hydrazide groups of compound **1** are characterized by two bands at 1707 and 1659 cm^−1^ [52]. In the cases of *N*-substituted compounds **3**, **5**, and **6**, pronounced bands were found at 1664, 1662, and 1672 cm^−1^, respectively. By contrast, *N*-substituted compounds **2** and **4** had corresponding IR bands shifted by 1632 and 1630 cm^−1^, respectively. These bands are absent in the FTIR spectrum of compound **7**, in which hydrazide groups are replaced by methyl esters (CO–O–CH_3_), which absorb at 1711 cm^−1^.

### 2.3. Binding Study of Compounds ***1***–***6*** with Fe(II) Ions

The interaction of prepared compounds **1–6** with Fe(II) ions was determined by absorption spectroscopy, and the highest affinity (represented by the binding constant, Table 1) was obtained for compound **1** (Figure 3 and Table 1). Its value suggests that chelation of Fe(II) ion could be associated with its potentially biological effects. On the other hand, compounds **3**, **5**, and **6** showed no significant interaction with Fe(II) ions. These results showed that the hydrazine group is very important for the chelation of Fe(II) ions by pyrrolo[3,2-*b*]pyrrole chelators. On the contrary, we expected that an incorporation of another iron-binding group (i.e., hydroxyphenyl) into the structural motif of compound **1** would support its chelation ability. Loss of chelation ability could be explained by the protonation of the iron-binding group. Nevertheless, pKas (Table 1) suggest that tested compounds are not positively charged at the relevant physiological pH. On the contrary, they can display a negative charge in the basic pH. On the other hand, the calculated pKa values suggest that compounds **1**, **2**, and **4** could not lose Fe(II) binding affinity at relevant cellular pH levels.

### 2.4. Inhibition of the TET1 Protein by Compounds ***1***–***6***

In the experiment presented, the inhibitory effect for TET1 was observed only for compound **1** with an IC_50_ value of 1.33 μM. These results suggest that the structural motif of the pyrrolo[3,2-*b*]pyrroles could be used in the design of TET1 protein inhibitors. Figure 4 and Appendix A show the dependence of TET1 activity on the concentration of compound **1**. Compounds **2**–**6**, which were screened under the same conditions, showed no inhibitory activity. Although compounds **2** and **4** displayed affinity for Fe(II), they exhibited no inhibitory effect towards TET1.

This inhibition plot implies, that compound **1** can be a potent inhibitor of the TET1 protein. On the other hand, the question arises as to whether the inhibition effect of compound **1** is not mainly caused by the chelation and consequent decrease in the level of free Fe(II) ions. There are two significant arguments against this interpretation.

First, the enzyme assay was performed in the presence FeSO_4_ (0.1 mM; TET1 protein cofactor). Assuming that the effect of compound **1** is caused by chelation of Fe(II), its 1.37 μM concentration (IC_50_) could decrease the level of free iron only by 1.37% without a significant effect on the enzyme activity.

Second, compounds **2** and **4** are also strong iron chelators, but despite that, no significant inhibition of TET1 activity was observed. It strongly implies that the inhibition by compound **1** is not dependent upon the chelation of free Fe(II) ions. In the case of the TET1 protein, a similar inhibition mechanism was also found for the other Fe(II) chelators (e.g., cannabinoids and Schiff bases) [32,33].

Accordingly, we assume that the mechanism of inhibition with compound **1** not only is based on affinity to Fe(II) but also could be influenced by the 3D structure and charge density of pyrrolo[3,2-*b*]pyrrole compounds that, in turn, affect TET1 protein binding. The substitution of chelator **1** with bulky phenyl groups probably reduced the affinity of compounds **2** and **4** for the Fe(II) ions by blocking effective uptake to the active site and, thus, reducing the inhibition of the TET1 protein.

### 2.5. Affinity of Compound ***1*** to TET1 Protein 1

To demonstrate binding properties of compound **1** (i.e., ligand) to the TET1 protein, we determined K_D_ using microscale thermophoresis. We observed a ligand concentration-dependent decrease in labeled protein fluorescence (Figure 5), and fluorescence measurements during serial dilution were performed to discriminate between binding-specific fluorescence quenching (Figure 6) and loss of fluorescence due to protein precipitation.

For this test, the remainder of tubes 1 to 3 and 14 to 16 prepared in the original binding assay were centrifuged for 10 min at 15,000× *g*. Thereafter, 10 μL of each sample was removed and mixed with an equal volume of SD-mix (4% SDS and 40 nM DTT (assay buffer)) and heated to 95 °C for 5 min to denature the protein. Subsequently, samples were loaded to standard capillaries, and the fluorescence was detected using a Monolith NT.115. We observed an identical fluorescence of bound and unbound state after denaturation, which indicates that the initial fluorescence loss was caused by binding-induced quenching (Figure 7). The K_D_ value was calculated from ligand concentration-dependent changes in normalized fluorescence (*F*_norm_) of the TET1 protein after 25 s of thermophoresis, based on the law of mass action using the NT. Affinity Analysis software. The experimentally determined dissociation constant (K_D_) was 27.2 ± 0.9 μM. The K_D_ value of compound **1** suggests that it may directly bind to the active site of the enzyme, which contributes to its inhibitory effect towards TET1. Nonetheless, K_D_ and IC_50_ values could not be directly compared as the K_D_ parameter determines the binding event and IC_50_ describes functional activity.

### 2.6. In Silico Docking of Compounds ***1***–***7*** to the TET1 Model

For the docking calculation, we used a structural model of the catalytic domain of TET1 (obtained for this work in a manner essentially identical to that published by Chua et al., 2019, starting from the 3D structure of TET2 (PDB id 4NM6) [45]. In silico mutations (from the amino acid sequence of TET2 to that of TET1), using the published sequence alignment, provided the supplemental information section [45]. Model adjustments were then carried out using the program Coot [53]. The resulting TET1 model contains the polypeptide chain and four ions (three Zn ions and one Fe ion). Compounds **1**–**7** (Figure 8 and Appendix A) were docked to the TET1 structural model using the AutoDock Vina suite of programs [54]. The values of binding energy are shown in Table 2. The computed value of the free energy of binding between compound **1** and TET1, −11.9 kcal/mol, which suggests a strong affinity of compound **1** for the TET1 protein and a potent inhibition activity towards this enzyme. The docking model predicts that compound **1** forms a conventional hydrogen bond with Cys1263, Thre1372, and His1257 and additional carbon–hydrogen bonds with His1386, Thre1372, Lys1269, and His1257.

This phenomenon could confirm the relevance of our hypothesis, that is, that the inhibition effect of compound **1** is associated with its affinity for Fe(II) ions. Accordingly, the substitution of the hydrazine group led to loss of inhibition. Nevertheless, functional groups of compound **1**, including hydrazine, can also interact with residues of TET1 proteins via hydrogen bonds and hydrophobic interactions.

A further explanation could be due to larger sizes of compounds **2**–**6** against compound **1**, producing steric hindrance. The computed values of binding energy between docked compounds **1**–**6** and the TET1 protein suggest that the affinity of compounds **2**–**6** for the protein is significantly decreased compared with smaller compound **1**. We cannot exclude that they are too bulky and that their possible binding in the enzyme cavity is inadequate for effective enzyme inhibition. As suggested with the above docking compounds **2**–**6** to the TET1 protein (Appendix A), the hydrazine groups (iron chelation moiety) are outside of the cavity, and their Fe(II) interaction/chelation is not possible. In comparison, the binding energy of compound **7** is −5.34 kcal/mol. This value is higher than the calculated interaction energy for compound **2** (−3.97 kcal/mol); however, the binding modes of compound **7** (Appendix A) showed a significantly greater distance to Fe(II) ions than in the case of compounds **2**–**6** (Appendix A).

The presented model could also explain why the Fe(II) binding affinity of compounds **1**–**6** (represented by binding constants in Table 1) did not correlate with their binding energies for the TET1 protein. Docking studies showed that only in the case of compound **1** can the hydrazine group binding in the enzyme cavity accommodate interaction with the enzyme Fe(II) ion. Additionally, compounds **2**–**6** (fully) and compound **1** (partially) interact with amino acid residues of TET1 proteins. This suggests, in the case of compounds **2**–**6,** that their interaction energies with the TET1 protein are most probably not dependent on their Fe(II) binding affinity.

It is suggested from the above data and discussion that the hydrazine groups of compound **1** play an important role in enzyme inhibition, and their binding to the Fe(II) ion within the TET1 protein is part of the inhibition mechanism.

### 2.7. Influence of the Compounds ***1***–***6*** on the Cell Viability

The effects of the tested compounds **1**–**6** on HF-P4, U-118 MG, H1299, BT-20, and Caov-3 cell lines were evaluated by the MTT assay during 48 h treatment (Table 2 and Appendix A). In all tested compounds, we recognized a difference in cytotoxicity between healthy cells (HF-P4) and cancer cell lines (BT-20, Caov-3, and H-1299). The greatest cytotoxic effect was observed for compounds **1** and **4**. The presence of the NH_2_ group in compound **1** and one more OH group in compound **4** indicates that a higher polarity functionality can increase the cytotoxic effect. Obtained values of IC_50_ (Figure 9 and Table 3) suggest that the inhibition of the TET1 protein could have promising potential in anticancer treatment. For example, Filipczak et al. recently published that the NSCLC cell line (e.g., H-1299) displayed increased expression of TET1 [23]. Its inactivation by si-RNA led to DNA single- and double-strand breaks and induction of p21, which resulted in cell senescence and genomic instability in p53-mutant cell lines. In the case of ovarian cancer, the authors discovered oncogenic and poor prognosis roles of TET1 in epithelial ovarian carcinoma (EOC). TET1 is believed to demethylate the epigenome and activate numerous oncogenic pathways in EOC. The aforementioned study presents a new mechanism where TET1 contributes to tumor development through collaboration with CK2α [21]. The overexpression of the TET1 protein was also found in triple negative breast cancer and was associated with an activation of PI3K/mTOR pathway and decreased overall survival [19].

Although only compound **1** inhibited the TET1 protein, significant cytotoxic effects were also observed for other tested compounds **2**–**6**. We cannot overlook that hydrazide derivatives can display various cytotoxicity effects [38,39,40]. On the other hand, the higher cytotoxic effects in cancer cells was observed only in the case of compound **4** for H-1299 and U-118 cell lines. Nevertheless, compound **4** displayed high cytotoxicity for control healthy HF-P4h cells.

Compounds **1**, **2**, and **4** also display strong chelation ability for Fe(II) ions, and their effect on iron homeostasis cannot be neglected. We suggest that their concentration could also lead to decreased activity of other Fe(II)-dependent enzymes, such as TET proteins. For example, in HEK293 cells, the application of iron chelators led to increased global DNA methylation associated with a decrease in TET1 activity in [43]. Nevertheless, the intracellular level of free iron ions varies from submicromolar to micromolar concentrations, depending upon the cell type, external conditions, and methods used [55].

### 2.8. Intracellular Distribution of Compounds ***1***–***6***

Using real-time live-cell fluorescence microscopy on the model HF-P4 cell line, we investigated the intracellular localization of tested compounds at a concentration of 1 μM. To prove the specific intracellular localization of compounds, we used the organelle fluorescent trackers MitoTracker^®^ Red FM and LysoTracker^®^ Green FM (Figure 10). Visualization of compounds **3**, **4**, and **6** suggested their intracellular localization in lysosomes, which was confirmed by calculating Pearson’s coefficient for the colocalization (Figure 11). Compounds **2** and **5** were localized in both mitochondria and lysosomes. For compound **1**, we observed an intracellular structure resembling a mitochondrion, which was further confirmed by a positive linear relationship between MTR and compound **1** and proved the selectivity for mitochondria.

Among compounds with a purely lysosomal localization (compounds **3**, **4**, and **6**), compound **4** displayed the highest cytotoxicity for all tested cell lines. Unlike compounds **3** and **6**, compound **4** has strong affinity for the Fe(II) ion. It suggests that the cytotoxic effect of compound **3** is at least partially connected with iron chelation. It is well known that lysosomes contain a considerable amount of redox-active iron due to the high degradation of endocytosed and autophagocytosed iron-containing molecules and organelles (e.g., mitochondria) [56]. It is assumed that lysosome and lysosome-related organelles play essential roles in the dynamic vesicle-mediated trafficking of iron.

Dual localization of compounds **2** and **5** suggest that that either of these compounds could independently target both mitochondria and lysosomes. Alternatively, the aforementioned distribution pattern could be suggesting that their cytotoxic mechanisms could be associated with the induction of mitophagy (mitochondrial autophagy). Mitochondrial damage and the loss of mitochondrial membranes probably induced autophagosome formation and lysosome degradation of mitochondria [57,58,59,60].

Compound **1** surprisingly displays exclusive mitochondrial localization and potent inhibition activity against the TET1 protein. Hydrophobic compounds, which display cationic charge, can be localized in the mitochondria due to mitochondrial membrane potential [61]. The question is whether this mechanism can be applied in the case of compound **1**. Its interaction with Fe(II) and its inhibition effect against the TET1 protein were studied at neutral pH. Functional binding groups for cationic analytes, such as Fe(II), require high electron density, and H^+^ bonding/interaction leading to a possible cationic charge of compound **1** would strongly repress its chelation and inhibition activity due to the interaction of the Fe(II) ion as part of the inhibition mechanism. It is practically impossible that the pyrrolo[3,2-*b*]pyrrole scaffold could display a cationic charge, at least at physiological pH. The pH of the mitochondrial matrix is weakly alkaline (approximately 7.6–8.0), and thus, mitochondrial uptake cannot lead to protonation of compound **1** [62]. Additionally, analysis of its chemical structure by the ChemDraw program (20.0) did not detect any protonatable group. Nevertheless, we proved that compound **1** is a potent Fe(II) chelator. Chelated iron displayed its cationic charge, and the resulting complex can be transported into mitochondria. As implied from the above, the intramitochondrial concentration of compound **1** could be higher sometimes than the administered concentration. On the other hand, the derivatization of compound **1** in the cell cannot be ruled out; especially, the oxidative mitochondrial environment [63] could also modulate its cellular localization and cytotoxic effects.

Similar results were observed for compounds **2**–**6**, although compounds **3**–**5** could display a negative charge in the pH milieu of the mitochondrial matrix. On the other hand, this phenomenon could partially explain why compound **4** with affinity for the Fe (II) ion displays mitochondrial co/localization as with compounds **2** and **1**.

Exclusive mitochondrial localization and potent inhibition activity against the TET1 protein suggests that compound **1** could represent a promising structural motif for targeting the mitochondrial epigenome. At the present time, the most used and developed epigenetic agents are focused on chromosomal targeting. Nevertheless, the relevance of the mitochondrial epigenome is increasing [25]. Epigenetic mechanisms are considered to be inheritable and impact both nuclear and mitochondrial DNA, but knowledge of the mitochondrial epigenome is limited, and many aspects of the epigenetic regulation of mtDNA remain unclear. Various high-impact studies have shown that the mitochondrial epigenome and the level of hydroxymethyled cytosine may be important in key regulatory processes of the cell cycle and the pathogeneses of numerous serious diseases, such as cancer [26,64,65,66,67,68]. In addition, the presence of TET proteins in mitochondria, including the TET1 protein, was reported [27,28]. More importantly, decreased levels of the mitochondrial TET1 protein lead to a decrease in 5hmC content [27]. Specific inhibition of the mitochondrial TET1 protein could be a promising therapeutic tool for the regulation of the 5hmC content in the mitochondrial DNA.

There are, nevertheless, several challenging aspects. First, compound **1** can have other cytotoxic effects other than the inhibition of TET1 proteins. It can be neglected that compound **1** is a potent Fe(II) chelator. It can be expected that the intramitochondrial level of compound **1** can be higher than the applied dose (due to its mitochondrial accumulation), and consequently, the intramitochondrial iron pool would decrease [69]. For example, significant decreases in the mitochondrial free-iron level can induce mitophagy. However, compound **1**, unlike compounds **2** and **5**, does not display dual lysosome and mitochondrial localization, which, in the case of mitochondria being engulfed by the lysosome, would be a strongly expected phenomenon. On the other hand, at least in part, compound **1** is distributed in the form of an iron complex. A decrease in the level of the mitochondrial free iron could be mitigated.

Second, compound **1** could also decrease the activity of the nuclear TET1 protein. Although we observed exclusive mitochondrial localization of compound **1**, its unobserved nuclear localization cannot be ruled out. Nevertheless, based on our observation and proposed mechanism, we can expect that compound **1** will significantly affect the mitochondrial epigenome, rather than the nuclear ones. It is true that cationically charged compounds can also be localized in the nucleus [61]. Nevertheless, they are significantly less hydrophobic than compound **1**, including its iron complex. Additionally, nuclear DNA contains billons of base pairs, and although cells contain hundreds to thousands of copies of mitochondrial DNA, their numbers of base pairs is several orders of magnitude smaller than nuclear DNA [70]. This could suggest that the amount of nuclear TET1 proteins is significantly higher than in mitochondria, and the effective targeting of mitochondrial TET1 could require significantly less inhibitor. On the other hand, we cannot exclude that targeting the mitochondrial TET1 protein could dysregulate mitochondrial metabolism, such as the citric acid (Krebs) cycle. It is well known that some metabolites of the Krebs cycle are TET1 protein inhibitors (fumarate and succinate) and activators (2-oxoglutarate) [71,72].

Third, the inhibition of the TET1 protein may not have the expected effect on gene activity. For example, it was reported that mutant TET1 protein—without enzymatic activity—could modulate gene expression, similar to native proteins [73]. In addition, it cannot be excluded that TET2 and TET3 proteins can also catalyze the hydroxymethylation of the methylcytosine groups, similar to the TET1 protein. Nevertheless, the Chen et al. high-impact studies strongly suggest that the TET1 protein could play a major role in cytosine hydroxymethylation [27]. On the other hand, it was reported that, unlike in nuclear DNA, the majority of 5mC and 5hmC were not found in the CpG islands [74]. In this context, we should mention that short isoforms of the TET1 protein cannot contain the CXXC domain for the recognition of CpGs [75]. The representation of the TET1 protein isoform in mitochondria is not widely known. Besides, high hydroxymethyl and methyl cytosine levels were also found in the D-loop, the region of mitochondrial DNA responsible for its replication and transcription [29,74,76].

Fourth, the TET1 inhibition activity of compound **1**, as represented by the IC_50_ values, appears to be lower than expected for a “promising” drug candidate. For example, in this case of the hydrazone chelator, the reported IC_50_ value (0.79 μM) was significantly lower than that determined for compound **1** [33]. This means that the effective inhibitory concentration of compound **1** should be at least a micromolar concentration. However, the mechanism of its mitochondrial localization (discussed above) can sometimes increase its mitochondrial concentration and thereby bewitch its inhibition activity.

Fifth, regarding the above mentioned, it is difficult to predict the effect of TET1 protein inhibitors on mitochondrial metabolism and cell viability. It could be expected that this type of agents could modulate mitochondrial metabolism and activate the mitochondrial apoptotic pathway. On the other hand, it cannot be overlooked that the cytotoxicity of compound **1** for the tested cancer lines (represented by IC_50_) is at best in a micromolar concentration. However, although inhibition was not associated with significant cytotoxicity, mitochondria also play a significant role in cancer metastasis and recurrence [77]. Furthermore, studying the role of mitochondrial TET1 and mitochondrial DNA hydroxymethylation may not be so difficult with the use of specific inhibitors against the mitochondrial TET1 protein. Finally, however, the possible therapeutic potential of a TET **1** protein inhibitor with mitochondrial selectivity, such as compound **1,** must be validated by other studies.

## 3. Materials and Methods

All chemicals and solvents were purchased from commercial vendors (TCI and Merck/Sigma-Aldrich, Prague, Czech Republic) at the quoted purity and used without additional purification, if not mentioned otherwise. Nuclear magnetic resonance (NMR) spectra were recorded on a 500 MHz instrument (Bruker BioSpin Ettlingen, Germany) at room temperature in DMSO-d6. The chemical shifts (δ) are presented in ppm, coupling constants (J) in Hz. The program MestReNova v 14.2 was used to process the NMR spectra. The UV–VIS absorption spectra were recorded using a Varian Cary 400 SCAN UV–VIS spectrophotometer (Varian, Palo Alto, CA, USA), in which the reference spectrum of plain solvent was subtracted from all sample spectra. High-resolution mass spectrometry (HRMS) spectra were obtained using electrospray ionization (ESI) with an LTQ Orbitrap spectrometer. All compounds are >95% pure by elementary analysis.

### 3.1. Synthesis

#### 3.1.1. Diester **7**

A mixture of methyl 4-formylbenzoate (1 g, 6.1 mmol), *p*-toluidine (0.65 g, 6.1 mmol), and *p*-toluenesulfonic acid (110 mg, 0.6 mmol) in glacial acetic acid (5 mL) was heated at 90 °C for 30 min. To the thick yellow mixture, 2,3-butanedione (260 μL, 2.9 mmol) was added, and the mixture was heated at 90 °C for 3 h. The reaction mixture was then cooled to room temperature. The yellow precipitate was filtered, washed with glacial acetic acid, and recrystallized by dichloromethane/hexane (1:1) to produce diester **7** as yellow solid (0.51 g, 15%). ^1^H NMR (DMSO-d_6_): δ 7.75 (d, J = 7.9 Hz, 4H), 7.25 (m, 2H), 7.15 (d, J = 7.9 Hz, 2H), 7.10–7.02 (m, 8 H), 6.35 (s, 2H), 3.77 (s, 6H), 2.27 (s, 6H) ppm. ^13^C NMR (DMSOd_6_): δ 165.73, 137.19, 137.09, 135.89, 135.20, 132.56, 130.79, 130.35, 129.49, 127.39, 125.28, 95.82, 52.33, 20.91 ppm. mp: 317–319 °C. Elem. Anal. Calcd for C_36_H_30_N_2_O_4_: C, 77.96; H, 5.45; N, 5.05. Found: C, 77.93; H, 5.48; N, 5.03. HRMS (ESI^+^) for C_36_H_30_N_2_O_4_ [M + H]^+^ calculated: 555.22783, found: 555.22816.

#### 3.1.2. Compound **1**

A mixture of diester **7** (500 mg, 0.90 mmol) and a catalytic amount of 4-(dimethylamino)pyridine (11 mg, 0.09 mmol) in hydrazine monohydrate (9 mL) was heated under reflux for 2 days. After cooling, the precipitate was filtered and washed with dichloromethane to remove unreacted starting material. The precipitate was recrystallized by dichloromethane to obtain compound **1** as yellow solid (300 mg, 60%). ^1^H NMR (DMSO-d_6_): δ 9.70 (s, 2H), 7.74–7.62 (m, 4H), 7.45–7.08 (m, 12H), 6.55 (s, 2H), 4.57 (bs, 4H), 2.43 (s, 6H) ppm. ^13^C NMR (DMSOd_6_): δ 165.83, 137.29, 137.24, 135.99, 135.30, 132.66, 130.89, 130.45, 127.49, 127.39, 125.38, 95.92, 21.01 ppm. mp: 312–313 °C. Elem. Anal. Calcd for C_34_H_30_N_6_O_2_: C, 73.63; H, 5.45; N, 15.15. Found: C, 73.60; H, 5.48; N, 15.11. HRMS (ESI^+^) for C_34_H_30_N_6_O_2_ [M + H]^+^ calculated: 555.25030, found: 555.25018.

#### 3.1.3. General Procedure for the Synthesis of Compounds **2**–**6**

Compounds **2**–**6** were synthesized by the reaction of hydrazine **2** with corresponding substituted salicylaldehydes. Hydrazine **1** (20 mg, 0.036 mmol) in 1 mL ethanol reacted with excess of corresponding aldehyde (0.36 mmol) in the presence of 10 mol% acetic acid at room temperature for 24 h. The reaction mixture then evaporated to dryness, the residue was suspended in dichloromethane, and the yellow precipitate was filtered and washed with DCM till removing all excess of aldehyde to get compounds **2**–**6** as yellow solid (70–90% yield).

#### 3.1.4. Compound **2** (82% Yield)

^1^H NMR (DMSO-d_6_): δ 11.76 (s, 2 H), 11.46 (s, 2H), 8.41 (s, 2H), 7.79 (d, J = 8.12 Hz, 4H), 7.38–7.26 (m, 8H), 7.24–7.12 (m, 6H), 6.59 (s, 2H), 6.27 (d, J = 8.27 Hz, 2H), 6.13 (s, 2H), 3.37 (q, J = 7.11 Hz, 8H), 2.37 (s, 6H), 1.11 (t, J = 6.95 Hz, 12H) ppm. ^13^C NMR (DMSOd_6_): δ 163.83, 162.11, 160.12, 154.24, 151.14, 150.18, 137.28, 136.52, 136.17, 135.42, 132.98, 132.05, 130.54, 127.96, 127.52, 125.54, 104.92, 96.35, 96.06, 44.57, 21.03, 12.96 ppm. mp: 306–307 °C. Elem. Anal. Calcd for C_56_H_56_N_8_O_4_: C, 74.31; H, 6.24; N, 12.38. Found: C, 74.24; H, 6.26; N, 12.34. HRMS (ESI^+^) for C_56_H_56_N_8_O_4_ [M + H]^+^ calculated: 905.44973, found: 905.44958.

#### 3.1.5. Compound **3** (79% Yield)

^1^H NMR (DMSO-d_6_): δ 12.11 (s, 2 H), 11.03 (s, 2H), 8.61 (s, 2H), 7.86–7.78 (m, 4H), 7.48–7.10 (m, 16H), 6.98–6.92 (m, 2H), 6.61 (s, 2H), 2.37 (s, 6H) ppm. ^13^C NMR (DMSOd_6_): δ 162.31, 156.07, 154.52, 153.56, 145.98, 136.80, 135.73, 134.96, 132.61, 130.07, 129.72, 129.13, 127.71, 127.06, 125.09, 119.77, 117.55, 113.96, 95.96, 20.56 ppm. mp: 337–338 °C. Elem. Anal. Calcd for C_48_H_36_F_2_N_6_O_4_: C, 72.17; H, 4.54; N, 10.52. Found: C, 72.13; H, 4.58; N, 10.48. HRMS (ESI^+^) for C_48_H_36_F_2_N_6_O_4_ [M + Na]^+^ calculated: 821.26583, found: 821.26505.

#### 3.1.6. Compound **4** (81% Yield)

^1^H NMR (DMSO-d_6_): δ 11.83 (s, 2H), 11.45 (s, 2H), 9.95 (s, 2H), 8.43 (s, 2H), 7.79–7.76 (m, 4H), 7.34–7.12 (m, 14H), 6.60 (m, 2H), 6.35–6.35 (m, 2H), 6.27 (s, 2H), 2.32 (s, 6H) ppm. ^13^C NMR (DMSOd_6_): δ 162.35, 161.13, 159.92, 149.41, 137.28, 136.65, 136.17, 135.42, 133.01, 131.78, 130.53, 130.42, 128.04, 127.52, 125.54, 110.98, 108.12, 103.11, 96.10, 21.03 ppm. mp: 328–329 °C. Elem. Anal. Calcd for C_48_H_38_N_6_O_6_: C, 72.53; H, 4.82; N, 10.57. Found: C, 72.53; H, 4.85; N, 10.53. HRMS (ESI^+^) for C_48_H_38_N_6_O_6_ [M + Na]^+^ calculated: 817.27450, found: 817.27402.

#### 3.1.7. Compound **5** (72% Yield)

^1^H NMR (DMSO-d_6_): δ 12.04 (s, 2H), 11.00 (s, 2H), 8.63 (s, 2H), 7.84–7.81 (m, 4H), 7.39–7.17 (m, 12H), 7.14 (d, J = 7.82 Hz, 2H), 7.04 (d, J = 7.99 Hz, 2H), 6.87 (t, J = 7.89 Hz, 2H), 6.61 (s, 2H), 3.82 (s, 6H), 2.37 (s, 6H) ppm. ^13^C NMR (DMSOd_6_): δ 162.65, 148.38, 147.58, 137.28, 136.79, 136.20, 135.41, 133.06, 130.55, 130.29, 128.15, 127.54, 125.56, 124.97, 121.26, 119.46, 119.37, 114.23, 96.16, 56.27, 21.04 ppm. mp: 321–322 °C. Elem. Anal. Calcd for C_50_H_42_N_6_O_6_: C, 72.98; H, 5.14; N, 10.21. Found: C, 72.93; H, 5.13; N, 10.23. HRMS (ESI^+^) for C_50_H_42_N_6_O_6_ [M + Na]^+^ calculated: 845.30580, found: 845.30522.

#### 3.1.8. Compound **6** (89% Yield)

^1^H NMR (DMSO-d_6_): δ 12.06 (s, 2H), 11.31 (s, 2H), 8.62 (s, 2H), 8.11–8.04 (m, 2H), 7.88–7.76 (m, 4H), 7.59–7.48 (m, 2H), 7.39–7.13 (m, 12H), 7.04–6.90 (m, 4H), 6.61 (s, 2H), 2.37 (s, 6H) ppm. ^13^C NMR (DMSOd_6_): δ 162.21, 157.42, 148.02, 136.78, 136.32, 135.72, 132.59, 131.29, 130.06, 129.73, 129.11, 127.66, 127.05, 126.66, 125.05, 119.27, 118.64, 116.37, 95.67, 20.54 ppm. mp: 332–333 °C. Elem. Anal. Calcd for C_48_H_38_N_6_O_4_: C, 75.57; H, 5.02; N, 11.02. Found: C, 75.51; H, 5.04; N, 11.00. HRMS (ESI^+^) for C_48_H_38_N_6_O_4_ [M + Na]^+^ calculated: 785.28467, found: 785.28415.

### 3.2. Vibrational Spectroscopy

FTIR spectra of compounds **1**–**7** were recorded on a Nicolet iS50 FTIR spectrometer using the ATR accessory (ZnSe crystal) with a resolution of 4 cm^−1^ in the range of 4000–400 cm^−1^. Three hundred and twenty scans were collected for each spectrum. All spectra were corrected for the carbon dioxide and humidity in the optical path. Smoothing, ATR, and baseline corrections were performed using the Omnic 8.2 software. Raman spectra (region 400–2000 cm^−1^, resolution 4.5 cm^−1^) of compounds **1**–**7** were recorded on a B&W Tek i-Raman Plus spectrometer equipped with diode laser (λ_ex_ = 785 nm, power 90 mW, which corresponds to 20% of the maximal power of 450 mW). Each spectrum was generated by the accumulation of 20 scans with a laser exposure time of 20 s per one scan at 22 °C. Smoothing and baseline correction were performed using the Origin 8 software.

### 3.3. Determination of Conditional Binding Constants and Complex

The complexation of compounds **1**–**6** with Fe(II) ions was studied by UV–VIS spectroscopy in aqueous medium (water/DMSO, 99:1, *v*/*v*). Compounds **1**–**6** were dissolved in DMSO (Penta, Katovice, Czech Republic) that, prior to the experiment, had been diluted with water to a concentration of 10 μM. The concentrations of Fe(II) ions varied from 0 to 0.5 mM. All titrations were performed in the same environment, and the ratio of DMSO to water was held constant. All absorption spectra of the chelators were recorded by a Cintra 404 GBC spectrometer. Conditional constants (Ks) were calculated from the absorbance changes, ΔA, of compounds **1**–**6** at the spectral maxima of their complexes with Fe(II), using nonlinear regression analysis with the Letagrop Spefo 2.0 (Vanura, Prague, Czech Republic) software. The computational model for conditional constants (Ka) is described and discussed in detail elsewhere [78], with the same method used to study the interactions of the organic hosts with metal ions in aqueous medium [79,80,81,82]. Solutions were mixed in the cuvette during the whole titration and after every Fe(II) chelate addition.

### 3.4. Inhibition of TET1 Protein

Compounds **1**–**6** were tested for the inhibition of the total 5mC hydroxylase TET enzyme using a purified TET1 enzyme (Active Motif, Carlsbad, CA, USA) by a fluorometric TET hydroxylase activity quantification kit (Abcam, Cambridge, UK). A total of 25 µg of purified TET1 protein was diluted in a TET assay buffer in a 5 mL volumetric flask, containing α-ketoglutarate and ferric sulfate. The concentrations of ascorbic acid, α-ketoglutarate, and ferric sulfate in the diluted TET assay buffer were 2, 1, and 0.1 mM, respectively. Tested compounds were dissolved in DMSO to obtain a concentration of 0.01 M. DMSO solutions of compounds **1**–**6** were then diluted with a TET assay buffer to concentrations of 10^−6^, 10^−5^, 10^−4^, and 10^−3^ M (in 10% of DMSO). Subsequently, 50 μL of the final TET assay buffer containing the TET1 protein and 5 μL of chelator’s solution was applied to each sample well. In control wells (TET1 without inhibitors), 5 μL of the final TET assay buffer with 10% DMSO was used instead of chelator’s solution. The final concentrations of chelators in a well plate were 10^−7^, 10^−6^, 10^−5^, 10^−4^, and 0 M with less than 1% of final concentration of DMSO. The following steps were performed according to manufacturer’s protocol. The TET1 protein converts methylated substrate to hydroxymethylated products, which could be recognized by a specific antibody. We determined the residual activity of the TET1 protein by measuring a fluorescence intensity on an enzyme-linked immunoabsorbent assay reader (ELISA), Infinite^®^ 200 PRO (Tecan, Switzerland, Switzerland), at an excitation wavelength at 530 nm and an emission wavelength at 590 nm. Each concentration was measured twice within one test, and each test was repeated three times. IC_50_ values of active inhibitors were determined using GraphPad Prism 8 via nonlinear regression.

### 3.5. Microscale Thermophoresis

Microscale thermophoresis (MST) is a sensitive method that can be used to assess biomolecular interactions and has been utilized to study interactions between a variety of binding partners of different molecular sizes [83]. MST was performed using a Monolith NT.115 instrument (NanoTemper Technologies GmbH, Munich, Germany). Binding affinity was measured between compound **1** at the 0.5 mM final concentration in HEPES (pH = 7.5) with 2.5% of DMSO and a recombinant His-tagged TET1 protein (Active Motif, Carlsbad, CA, USA). The His-tagged TET1 protein was labeled using an MO-L018 His-tag labeling kit, RED-tris-NTA, following the manufacturer’s protocol. A 16-step dilution was performed by adding 20 µL of compound **1** at 2× higher concentration (1 mM) to the first PCR tube and 10 µL of buffer (HEPES, pH = 7.5) to PCR tubes 2–16. In the next step, we transferred 10 µL of the ligand from PCR tube 1 to PCR tube 2 and mixed by pipetting up and down at least six times. We repeated this procedure, removing 10 μL after mixing from each of tubes 3–16 and discarding the last 10 µL from PCR tube 16. The first capillary contained the highest concentration of the ligand (e.g., compound **1**), and the 16th capillary contained the lowest concentration of the ligand. An amount of 10 µL of labeled protein was added to each PCR tube and mixed by pipetting up and down. The final concentration of the His-tagged TET1 protein was 50 nM. Samples from each PCR tube were loaded to NT.115 standard capillaries and inserted into the chip tray of the MST instrument for thermophoresis. Signals were recorded using 20% MST power and 40% LED power after 25 s. Measurement was repeated two times, and data were analyzed using the NTAnalysis software. Affinity is quantified by analyzing the change in *F*_norm_ as a function of the concentration of the titrated binding partner.
*F*_norm_ = (1 − *FB*)*F*_norm,unbound_ + (*FB*)*F*_norm,bound_

*FB*: fraction bound; 

*F*_norm,unbound_: normalized fluorescence of the unbound state; 

*F*_norm,bound_: normalized fluorescence of the bound state.

### 3.6. Docking to the TET1 Protein

To date, no high-resolution 3D structure of the catalytic domain of human TET1 has been deposited in the Protein Data Bank. Three entries for high-resolution structures of the iron-containing catalytic domain of human TET2 are available (PDB ids 4NM6, 5D9Y and 5DEU, L. Hu et al., 2013; L. Hu et al., 2015) [84,85]. To circumvent this lack, Chua et al. generated a 3D structural model of human TET1 starting from the high-resolution crystal structure of human TET2 (PDB entry 4NM6) with computational details given in the Appendix A section of their publication [45]. We followed the procedure of Chua et al. to generate a model of the catalytic domain of human TET1 using the published sequence alignment. In silico mutations and model adjustments were carried out using the program Coot [53]. The resulting model was subjected to limited energy minimization with the programs Phenix and Charmm [86,87]. The procedure was deemed appropriate, with a reduction of internal energy from 156871 to 149649 kcal/mol. The energy-minimized 3D model was then used for docking calculations. Docking was performed using AutoDock Vina [54]. The search was performed in a box at dimensions of 26 × 26 × 28 Å, centered at the entrance of the active site and located ca. 11.2 Å from the Fe(II) ion.

### 3.7. Cell Cultures

A human dermal fibroblast cell line was used for intracellular localization and cytotoxicity assay (MTT). Human fibroblast cells HF-P4 (human fibroblasts—patient 4) are normal dermal fibroblasts, which were obtained from the residual skin of a healthy patient who underwent plastic surgery (with informed consent of the patient). HF-P4 cell lines were obtained from the Department of Plastic Surgery, Third Faculty of Medicine, Charles University and University Hospital Královské Vinohrady, Prague, Czech Republic, as described by Szabo et al., 2011, and Dvořánková et al., 2019 [88,89], and have been processed in accordance with the relevant local ethics committee and the Declaration of Helsinki protection of patients’ rights and benefits with written and signed informed consent of each patient. Cells were cultivated under standard conditions in Dulbecco’s modified Eagle’s medium (ATCC, Manassas, VA, USA), supplemented with 10% FBS and streptomycin (100 μg/mL) at 37 °C at a 5% CO_2_ atmosphere.

A human glioblastoma cell line (U118-MG) and a human non-small-cell lung carcinoma cell line (H1299) were used for the cytotoxicity assay (MTT). Both cell lines were generously supplied by Dr. M. Masařik, PhD, Department of Pathological physiology, Masaryk University, Czech Republic. U118-MG cells were cultivated under standard conditions in Dulbecco’s modified Eagle’s medium (ATCC, Manassas, VA, USA) supplemented with 10% FBS and streptomycin (100 μg/mL) at 37 °C at a 5% CO_2_ atmosphere. H1299 cells were cultivated under standard conditions in RPMI 1640 (BioTech, Prague, Czech Republic) medium supplemented with 10% FBS and streptomycin (100 μg/mL) at 37 °C at a 5% CO_2_ atmosphere.

The human mammary gland/breast cell line (BT-20), purchased from ATCC (BT-20 (ATCC^®^ HTB-19™), Manassas, VA, USA), was used for the cytotoxicity assay (MTT), and BT-20 cells were cultivated under standard conditions in Eagle’s minimum essential medium (ATCC, Manassas, VA, USA) supplemented with 10% FBS and streptomycin (100 μg/mL) at 37 °C at a 5% CO_2_ atmosphere.

The human ovary adenocarcinoma cell line (Caov-3), purchased from ATCC (Caov-3 (ATCC^®^ HTB-75™), Manassas, VA, USA), was used for the cytotoxicity assay (MTT). Caov-3 cells were cultivated under standard conditions in Dulbecco’s modified Eagle’s medium (ATCC, Manassas, VA, USA) supplemented with 10% FBS and streptomycin (100 μg/mL) at 37 °C at a 5% CO_2_ atmosphere.

### 3.8. Determination of IC_50_ of Tested Compounds

For the cytotoxicity assay, we used the following cell lines: human glioblastoma cell line U118-MG, human mammary gland/breast cell line BT-20, human ovary adenocarcinoma cell line Caov-3, and non-small-cell lung carcinoma H-1299. As previously mentioned, these cell lines contain a high level of the TET1 protein, which corresponds to a lower overall survival. In addition to these cell cultures, we used the human dermal fibroblast cell line HF-P4 as a healthy control. Determination of cytotoxicity was performed using the MTT [3-(4,5-dimethylthiazol-2-yl)-2,5-diphenyltetrazolium bromide] colorimetric assay. Viable cells from each cell line were cultured in 96-well plates (VWR, Stříbrná Skalice, Czech Republic) at a density of 1.0 × 10^5^ cells per well and allowed to grow for 24 h in 200 μL of recommended medium. After a 1-day incubation, the medium was replaced with solutions of compounds **1**–**6** at concentration ranges of 0.1–20 μM. Each concentration was measured four times within one test, and the entire test was repeated three times. Tested compounds **1**–**6** were dissolved in DMSO and then diluted with corresponding medium to a final volume of 200 μL (maximum 1% DMSO). After 48 h exposure, cells were treated with 175 μL of MTT solution (P-LAB, Prague, Czech Republic) for 2 h. In the next step, MTT solution was removed, and 125 μL of DMSO (Penta, Prague, Czech Republic) was added to solubilize dark crystals of formazan formed in intact cells, and absorbance was measured at 570 nm by the enzyme-linked immunoabsorbent assay reader Infinite^®^ 200 PRO (Tecan, Männedorf, Switzerland). The final DMSO concentration was smaller than 1% (*v*/*v*). The absorbance of untreated cells was considered 100%, and IC_50_ values were calculated by the GraphPad Prism 8 software. As a parameter for cytotoxicity, we used a concentration that inhibits 50% of cell growth (IC_50_).

### 3.9. Intracellular Localization

The intracellular localization of tested compounds **1**–**6** was investigated by fluorescence real-time live-cell microscopy using a Leica TCS SP8 WLL SMD-FLIM microscope at 37 °C and a 5% CO_2_ atmosphere. The model cell line, HF-P4 cells, was seeded at a density of 4.0 × 104 per well on 22 × 22 mm glasses (VWR, Stříbrná Skalice, Czech Republic) in Petri dishes for live-cell imaging in a complete cell culture medium (DMEM with 1% streptomycin and 10% FBS), and cells were maintained overnight (24 h) for adhesion. After 24 h, the cells were incubated for 15 min (37 °C, 5% CO_2_) in a complete culture medium containing compounds **1**–**6** at 1 μM concentrations, together with 50 nM of MitoTracker^®^ Red FM (MTR; Thermo Fisher, Prague, Czech Republic) and 300 nM of LysoTracker^®^ Green FM (LTG; Thermo Fisher, Prague, Czech Republic). These results in the “model” cell line were used as standards for assessing the intracellular localization. Cells were then rinsed with PBS twice and left in a fresh medium without phenol red. We used a water objective, HC PL APO CS2 63× (NA 1.2), and laser with an excitation wavelength of 405 nm (10% power) with a fluorescence emission range 415–480 nm for visualization of cells. For MTR, we used a laser with an excitation wavelength of 579 nm (power 8%), and for LTG, we used a laser with an excitation wavelength of 504 nm (power 10%). Colocalization of the tested hydrazone derivates with mitochondria or lysosomes was assessed by correlation statistical analysis of the intensity values of red (MTR) or green (LTG) and blue (hydrazone derivate) pixels in dual-channel images. Pearson’s correlation coefficient was calculated using the ImageJ software.

## 4. Conclusions

In this study, we prepared a series of pyrrolo[3,2-*b*]pyrrole derivatives combining a fluorescent probe and an iron-binding group. Our aim was to characterize these compounds in terms of their Fe(II)-binding capacity, cytotoxicity with respect to cancer cell lines with high expression of the TET1 protein, intracellular distribution, and potential epigenetic effects. Compound **1** displayed a strong affinity for the TET1 protein, demonstrated as K_D_, and potent inhibition of this enzyme. The inhibition was proposedly based on the chelation of Fe(II) ion at the active site of the enzyme. Notably, its substitution by other iron binding groups (compounds **2**–**6**) led to a decrease in chelation capacity and the loss of the inhibition activity. Most likely, compounds **2**–**6** are too bulky, and their iron (II) binding groups prevent uptake into the enzyme cavity. In addition, compound **1** produced the most effective anticancer activity, although its cytotoxic effect towards the U118-MG cell line was significantly lower than for compound **4**. This phenomenon was associated with a change in intracellular localization from mitochondrial (i.e., compound **1**) to lysosomal (i.e., compound **4**). The present results imply that compound **1** could represent a promising structural motif for mitochondrial TET1 protein inhibitors.

## Data Availability

Not applicable.

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
