# Peer review of "Targeting of the Mitochondrial TET1 Protein by Pyrrolo[3,2-b]pyrrole Chelators"

_ijms, 2022, doi:10.3390/ijms231810850_

Round 1

Author Response

We would like to thank the reviewer for thorough reading and reviewing our manuscript and especially for her/his remarks that helped us to significantly improve our manuscript. We have taken the reviewer’s advice and comments into account carefully point-by-point and made the following changes and corrections in the revised manuscript:

The premise, that potential TET1 inhibitors might work by chelating Fe(II), is never tested by using known Fe chelators.  Instead, three of the new compounds chelate Fe but only Cpd 1 is an inhibitor. Computer docking suggests the bulkier molecules 2-6 may not access the active site, and that accessible hydrazines (only found in 1) may be essential to bind Fe(II) in the active site. It would seem that this key hypothesis is testable by enzyme binding studies.

            It is true, that chelation of free Fe(II) ions (TET1 cofactor) can lead significantly repress enzyme activity without direct interaction with enzyme. To avoid this mistake, enzyme inhibition was tested in the presence FeSO4 (0.1mM). IC50 determined for the compound 1 was 1.37 mM. If the inhibition activity was caused by mainly chelation by free Fe(II), decrease level of free iron could be a maximum of o 1.37% (assumed stoichiometry 1:1). But enzyme activity was reduced on the half at this concentration. That doesn't match at all possible decrease in the concentration of free iron.

Similarly inhibition mechanism we also found for the previously tested Fe(II) chelators (e.g., cannabinoids and Schiff bases) (ref. 39 and 40).

Beside other tested compound 2 and 4 are also potent iron chelators comparative with compound 1 with similarly structure, but despite that any significantly inhibition of TET1 protein was not observed. It is also strongly imply, that inhibition effect of compound 1 is most probably caused with direct interaction of enzyme.  

Above information was included and discussed in the current version of manuscript (pages 6-7, lines 213-225).

But there is a bigger problem. When added to culture cells, several of the compounds localize to mitochondria and/or lysosomes, but not the nucleus where most of the chromatin is. So the killing effects are attributed to possible mtDNA modifications.  Nothing is stated about mtDNA epigenetics and whether TET1 (the putative target) is known to be present in mitochondria.

     We agree with reviewer, that mitochondrial epigenetic is was mentioned to limited. In the current version of manuscript change introduction includes paragraph focused on short introduction of mitochondrial mechanism, especially hydromethylation of DNA, including role of the TET1 protein (ref. 27 and 28) (page 2, lines 75-86). Beside this problematic is discussed in the subchapter 2.8. (page 14 line 398-421).

Worse yet, all the compounds kill cells (normal and cancer) within roughly the same range of IC-50 (generally high uM), whether or not they inhibit TET1 in the test tube or localize to mitochondria. So the reader is left very confused, with one likely explanation that cell killing is due to effects other than inhibition of TET1, possibly involving Fe chelation but not requiring specific localization to mitochondrial or lysosomal processes. Considerable attention is given to autophagy but why pick that one process?

   Wee, agree with reviewer, that cytotoxicity tested compounds 1-6 was discussed to shortly. In the current version of manuscript is their cytotoxicity is discussed in the subchapter of 2.7. (page 11, 343-355) and in the point of view their intracellular localization is discussed in the subchapter 2.8. (pages 13-14 lines 384-397). In the shortly, we can say compounds 2-6 can display own cytotoxic effects not depend on direct inhibition of TET1 protein, for example chelation iron ions in the target organelles.

There are numerous other annoying faults in the paper, in addition to numerous minor errors in usage - and misspelling of methlycytosin(c)e in the abstract. In particular, the data for the fluorescence microscopy images (Figure 9) do not align with the Pearson correlation charts in Figure 10. Speculation in the Conclusion section about what these compounds might be useful for include several that are not supported by the data.  As stated this was a confusing paper to review.

Figure 10 (11 in the current version of manuscript) was corrected.

Manuscript was thoroughly corrected by native speaker.  

Conclusion was corrected and speculations was excluded.

Reviewer 2 Report

The authors presented their discovery of a series of pyrrolo-pyrrole chelators as TET1 inhibitors. They determined the compound 1 was a strong TET1 inhibitor with IC50 =1.33 uM and it exhibits potent anticancer activity. However, this paper requires major modifications before publication.

1.  The authors didn't explicitly present how they found the pyrrolo-pyrrole core structures that were presented in this manuscript. A graphic scheme to show the rationale/history of discovering this core structure might be very helpful and necessary.

2.  Section 2.1 and 2.2 has the same title "Synthesis of compounds 1-7", which must be modified.

3. Inconsistency between Compound-TET1 binding energies and their binding constants with Fe(II) ions, which will argue their hypothesis that the potency is attributed to the chelating to Fe(II). In Table1, we saw compound 2 has stronger Log(K) of 7.2 than compound 4 (6.9) and only slightly worse than compound 1 (8.4). However, int Table 2, the compound 2 has lowest binding energy of -3.97 kcal/mol, which is dramatically worse than compound 4 (-8.24 kcal/mol) and compound 1 (-11.9 kcal/mol).  A detailed explanation/discussion must be provided. 

 4. The authors hypothesized that the steric bulkiness might be the reason why bulky compound 2-6 are less potent than compound 1. Did the authors test the Fe(II) binding and TET1 binding for compound 7? Since compound 7 has the similar size as compound 1, it would be interesting to see the performance of compound 7.

Author Response

We would like to thank the reviewer for thorough reading and reviewing our manuscript and especially for her/his remarks that helped us to significantly improve our manuscript. We have taken the reviewer’s advice and comments into account carefully point-by-point and made the following changes and corrections in the revised manuscript:

  1. The authors didn't explicitly present how they found the pyrrolo-pyrrole core structures that were presented in this manuscript. A graphic scheme to show the rationale/history of discovering this core structure might be very helpful and necessary.

Rationale of used structure motive is described in the subchapter (page 3, lines 113-129) 2.1. Figure illustrated design of prepared compounds was be included into manuscript as Figure 1. We also added short description of design of pyrrolo-pyrrole based chelators (“We decided to combine strongly fluorescent pyrrolo-pyrrole core (fluorophore for intracellular visualisation) with suitable metal-chelating unit (bioactive part responsible for inhibition of TET1 enzyme).)”

  1. Section 2.1 and 2.2 has the same title "Synthesis of compounds 1-7", which must be modified.

It was typographic mistake, Correct name is FTIR and Raman analysis of compounds 1-7.

  1. Inconsistency between Compound-TET1 binding energies and their binding constants with Fe(II) ions, which will argue their hypothesis that the potency is attributed to the chelating to Fe(II). In Table1, we saw compound 2 has stronger Log(K) of 7.2 than compound 4 (6.9) and only slightly worse than compound 1 (8.4). However, int Table 2, the compound 2 has lowest binding energy of -3.97 kcal/mol, which is dramatically worse than compound 4 (-8.24 kcal/mol) and compound 1 (-11.9 kcal/mol).  A detailed explanation/discussion must be provided. 

It is true, that result from table 1 did not corelate with result from table 3. In the current version of manuscript we provided two possible explanation. Docking study showed, that compounds 2-6 are to bulky and they iron binding can not effectively uptake enzyme cavity and interact with Fe(II) ion. Tested compound 1-6 can also interact with other part enzyme such as amino acid residue and their affinity may not match their chelation ability. Information was included into manuscript (page 10, lines 307-313).

  1. The authors hypothesized that the steric bulkiness might be the reason why bulky compound 2-6 are less potent than compound 1. Did the authors test the Fe(II) binding and TET1 binding for compound 7? Since compound 7 has the similar size as compound 1, it would be interesting to see the performance of compound 7

It is good question. Unfortunately, chemical structure of compound 7 did not contain any iron chelation group. Because, in this case of compounds 3, 5 a 6 with same structure motif any affinity for the Fe(II) ions, Fe(II) was not observed.  Chelation of Fe(II) ion by compound 7 can not be expected.

Docking studies implies, that compounds 7 could interact with TET1 proteins. Calculated value of interaction energy was -5.24 kcal/mil (included into table 2) and shortly discussed in the subchapter 2.6 (page 10, line 303-306). It is true, that compound 2 display lover value of interaction energy, nevertheless found bind mode (included into supplementary information as Fig. S20) showed significantly higher distance of compound 7 to Fe(II) ion then in this case of compounds 2-6.

Round 2

Reviewer 1 Report

SEE ATTACHED FILE

Author Response

We would like to thank the reviewer for thorough reading and reviewing our manuscript and especially for her/his remarks that helped us to significantly improve our manuscript. We have taken the reviewer’s advice and comments into account carefully point-by-point and made the following changes and corrections in the revised manuscript:

The revised manuscript is improved but there are still problems with (1) the basic argument (logic) about what the compounds are doing and (2) poor usage and careless mistakes.

(1) The experimental work shows that Compound 1 inhibits TET1.  The in silico work suggests a mechanism and perhaps why the bulkier compounds do not inhibit.  It is possible that studying how compound 1 inhibits TET1 will give insights into enzyme mechanisms. However, all the other work is a separate story that detracts from this main line, with some poorly interpreted results and discussion. Let me try to explain:

(a) It looks like Compound 1 is taken up by mitochondria. That alone does not indicate that the enzyme targeted is mitochondrial since the level of uptake needed for mitochondria to fluoresce brightly is likely >> level bound (at high affinity) to the enzyme (wherever it is). In fact there is no evidence that Compound 1 binds to TET1 anywhere in the cell or that such binding is the mechanism that kills cells (as the authors note).  The molecule localizes to mitochondria because it is a membrane permeant weak base (cation), whose uptake is driven by the mitochondrial membrane potential.  Its concentration inside mitochondria is likely >> 1 uM used for incubation, and cell death could be associated with effects on ox phos, on mitochondrial TET (if it exists) or on some other mitochondrial or non-mitochondrial activity.

We agree with review, that hydrophobic compounds can be drive to mitochondria by own cationic charge. Nevertheless compound 1 cannot display cationic charge in the relevant physiological pH. Fe(II) chelation and TET1 inhibition was study in the neutral pH.  Functionality of iron binding group request high electron density, which is strongly repressed by H+ bonding. Mitochondrial pH is weakly alkaline (7.4-8 pH). Mitochondrial uptake of compound 1 most probably lead to a loss of positive charge and not to protonation of the molecule. In according with analysis of their chemical structure (Chem Draw) did not found any protonatable groups. Calculated pKa values was included into table 1 and they are discussed in the line 190-195. Nevertheless, we proved, that compound 1 is hydrophobic molecule with strong mitochondrial selectivity. Possible explanation of this paradox could be giving its strong affinity for the Fe(II) ion. Unlike alone compound 1, iron complex of compound 1 display cationic charge and can be effectively accumulated in the mitochondria. We cannot exclude, that above phenomena could cause strong increase in the concentration of the compound 1 in the mitochondria. We also agree, that its mitochondrial concentration will be some times higher, that than that which would correspond to the dose applied. Nevertheless, in the result, thank above mechanism (kindly recommended by reviewer), its mitochondrial inhibition activity will be strongly enchanted. More detailed discussed in the line 409-424

               On the other hand, although we observed exclusive mitochondrial localization of compound 1, its possible effect on the nuclear TET1 protein cannot be excluded. Nevertheless, we can give two points, which strongly could be strongly support significantly presented selectivity of compound 1 against mitochondrial TET1 protein.  Firstly, results obtained from the fluorescence microscopy suggest, that majority of applied dose is localized in the mitochondria. Although is true, that structure motif of nuclear probes also contains cationic charge, they are significantly more hydrophilic than compound 1, or its iron complex. Secondly, although is true, that cells can contain hundreds copy of mitochondrial DNA and only two copy nuclear DNA. Number of nuclear base pars is several orders of magnitude larger than in this case of mitochondrial DNA, it is strongly suggesting that dose of effectively targeting nuclear epigenetic mechanisms will be most probably significantly higher than in this case of mitochondrial ones. (line 455-469)

               We also agree with review, that other toxic effects of compound 1 could be also considered. Compound 1 displayed potent chelation ability against iron ions. We can expect, its application could be significantly decrease activity of iron depended enzyme, including respiration enzymes. On the other hand, mechanism of mitochondrial localization is based on its iron complexes which can at least partially mitigate decrease in the level of free mitochondrial iron. However, despite this we cannot excluded, that cytotoxic effect of the compound 1 could be significantly related with disturbance of mitochondrial respiration. (line 444-454)

Although compound 1 is potent inhibitor of TET1 protein and its localization in the mitochondria was confirmed. It is difficult to predict effect its inhibition on mitochondria, or cell viability. Inasmuch as mitochondrial DNA encodes of part of the respiration enzymes. Besides, hydroxymethyl and methyl cytosine was also found in the D-loop. It could be expected, that inhibition of mitochondrial TET1 protein can induce disturbance of mitochondrial respiration and maybe activation of mitochondrial apoptotic pathway. (line 470-488)

(b) The other five compounds (2-6) localize to lysosomes, driven by the pH gradient. Literature (e.g. on Nile Blue) indicates that fluorescent weak bases that label (localize to) lysosomes also can be taken up by mitochondria at higher concentrations (considered > 0.1 uM).  It seems unlikely that cell death from these compounds is related to that caused by Compound 1.  I will venture that many chemicals kill cells at concentrations > 10 uM.

We agree with review, that intracellular pH gradient generally play significantly role in intracellular distribution and its effect on the intracellular localization should be considered. Value calculated pKa of compounds 2-5 strongly suggest, that did not display positive charge in the relevant physiological pH. (discussed in the line 418-425) It also imply, that possible protonation of iron binding group cannot explain loss chelation activity of compounds 3, 5 and 6. (line 189-193) In the shortly, it is very probably, that their distribution of compounds 2-5 is not depended on the intracellular pH gradient.

It is true, that only compound 1 display potent inhibition against TET1 protein. On the hand, their possible similarity in their effects cannot be excluded. This topic is discussed in the manuscript line 359-365, 444-453 and 483 and 487.

Basically, the localization experiments would need to be titrated to be meaningful. The cell killing results are uninterpretable.  If anything, they suggest Compound 1 is NOT a good starting point for anticancer drug discovery, since it kills cancer cells for unknown reasons, and only modestly better than compounds that do not inhibit TET.

Unfortunately, compounds 1-6 was not primary designed as fluorescence probes. And their application in the lover concentration than 1 mmol/mol did not give observable fluorescence. On the other, hand higher concentration was toxic for cells.

We agree with reviewer, that prediction about usability in the compound 1, and or targeting mitochondrial TET1 protein in anticancer therapy still cannot be done yet. Nevertheless, our results strongly suggest that structure motif of compound 1 is good starting point for the development of selective inhibitor of mitochondrial TET proteins. However, possible therapeutic potential of this strategy muss be considered.

This is why I feel the paper (at least the second half) needs to be totally rewritten.

 Biological part of manuscript was rewritten, for example the better interpretation of obtained results, limitation is widely discussed, line 444-488.

(2) Examples re listed below of a few errors that seriously detract from the paper and can cause misunderstandings. There are more. If accepted the paper would need careful editing in its final form. 

Line 20) of pyrroloc(??)pyrrole derivatives

Line 41) In 2009, was first identified as a new cytosine modification

Line 77) Nevertheless, the above discuss practically only role 5hmC context in the nuclear DNA, or and TET1 protein level/activity

Lines 80) It assumed, high similarity between nuclear and mitochondrial epigenetic mechanisms, expect (except) histone modification

Line 81) For example, presence modified bases

Line 686) we have prepared a series of pyrrolo[3,2-b]pyrroles containing a fluorescent probe. (The compounds ARE themselves fluorescent.)

Line 693) Most likely compounds 2-6 are to (too) bulky and their iron (II) binding can uptake into enzyme cavity (I don't understand the meaning here)

The manuscript has been carefully and detailed reviewed in terms of English grammar and text fluency by native speaker Dr. Bettie Sue Masters (specialist in the biochemistry and microbiology).

Reviewer 2 Report

Thank you for modifying the manuscript. I suggest the acceptance.

Author Response

We would like to thank the reviewer for thorough reading and reviewing our manuscript and especially for her/his remarks that helped us to significantly improve our manuscript.

Round 3

Reviewer 1 Report

The authors have adequately addressed the key criticisms of the paper in the added material. I believe a re-organization would have resulted in a better, less confusing  paper but so be it.  A few additional notes:

1. The affinity of cpd 1 for TET1 seems lower than expected for a "promising" drug candidate. The same is true for toxicity assays. These large uM IC50s do not seem pharmacologically interesting. But I am not a pharmacologist. Perhaps comment on these points.

2. The fact that cpd 1 accumulates in the mitochondria against the (presumed) large negative membrane potential means that it or some derivative (produced in the cell) is positively charged. The authors might want to track this down. Not a problem with lysosomes, for which accumulation is driven only by pH gradient.  

3. Also, in future, the authors will need to work with isolated mitochondria to study any bioenergetic effects of cpd 1. Perhaps see early work on rhodamine 123 etc. I agree that local iron chelation is a likely culprit. But the possibilities are almost endless.

4. The new material is somewhat clumsily worded in places and would benefit from another editing for usage.

Author Response

We would like to thank the reviewer for thorough reading and reviewing our manuscript and especially for her/his remarks that helped us to significantly improve our manuscript. We have taken the reviewer’s advice and comments into account carefully point-by-point and made the following changes and corrections in the revised manuscript:

  1. The affinity of cpd 1 for TET1 seems lower than expected for a "promising" drug candidate. The same is true for toxicity assays. These large uM IC50s do not seem pharmacologically interesting. But I am not a pharmacologist. Perhaps comment on these points.

It is true, that ideally these values should be submicromolar. However, due to the mitochondrial accumulation of compound 1, its inhibitory potency will sometimes be higher for the mitochondrial TET1 protein. Discussed in the current version of the manuscript line 486-492.

In addition to the aforementioned apoptotic pathway, mitochondria play an important role in tumor biology in the formation of metastases and cancer recurrences. In addition, specific inhibitors of mitochondrial TET1 proteins may have contributed to explain the role of this enzyme in mitochondria and cells. Discussed in the current version of the manuscript line 496-502. 

  1. The fact that cpd 1 accumulates in the mitochondria against the (presumed) large negative membrane potential means that it or some derivative (produced in the cell) is positively charged. The authors might want to track this down. Not a problem with lysosomes, for which accumulation is driven only by pH gradient.

Of course, this phenomenon could also significantly affect cellular distribution of compound 1.  It can be expected that the chemical structure of compound 1 could be stable in the cytosol and its derivation would involve the mitochondria due to the mitochondrial oxidative environment. However, the study of this phenomenon is one of the topics of our future research.  This topic is given in the current version of manuscript, line 424-427.      

  1. Also, in future, the authors will need to work with isolated mitochondria to study any bioenergetic effects of cpd 1. Perhaps see early work on rhodamine 123 etc. I agree that local iron chelation is a likely culprit. But the possibilities are almost endless.

We agree. It is one of the planed studies for compound 1.

  1. The new material is somewhat clumsily worded in places and would benefit from another editing for usage.

                The manuscript has been extensively edited.